# MicroRNA-23a-3p Down-Regulation in Active Pulmonary Tuberculosis Patients with High Bacterial Burden Inhibits Mononuclear Cell Function and Phagocytosis through TLR4/TNF-α/TGF-β1/IL-10 Signaling via Targeting IRF1/SP1

**DOI:** 10.3390/ijms21228587

**Published:** 2020-11-14

**Authors:** Yung-Che Chen, Chiu Ping Lee, Chang-Chun Hsiao, Po-Yuan Hsu, Ting-Ya Wang, Chao-Chien Wu, Tung-Ying Chao, Sum-Yee Leung, Yu-Ping Chang, Meng-Chih Lin

**Affiliations:** 1Division of Pulmonary and Critical Care Medicine, Department of Medicine, Kaohsiung Chang Gung Memorial Hospital and Chang Gung University College of Medicine, Kaohsiung 83301, Taiwan; choupeen@gmail.com (C.P.L.); cchsiao@mail.cgu.edu.tw (C.-C.H.); hsupowan@yahoo.com.tw (P.-Y.H.); filling.tw@yahoo.com.tw (T.-Y.W.); my47104710@gmail.com (C.-C.W.); tychao@adm.cgmh.org.tw (T.-Y.C.); sumyeeleung@hotmail.com (S.-Y.L.); b9002087@cgmh.org.tw (Y.-P.C.); 2Department of Medicine, Chang Gung University, Taoyuan 33302, Taiwan; 3Graduate Institute of Clinical Medical Sciences, College of Medicine, Chang Gung University, Taoyuan 33302, Taiwan; 4Department of Medical Research, Kaohsiung Chang Gung Memorial Hospital and Chang Gung University College of Medicine, Kaohsiung 83301, Taiwan

**Keywords:** active pulmonary TB disease, latent TB infection, miR-23a-3p, IL10, TLR4

## Abstract

The aim of this study is to explore the role of microRNAs (miR)-21/23a/146a/150/155 targeting the toll-like receptor pathway in active tuberculosis (TB) disease and latent TB infection (LTBI). Gene expression levels of the five miRs and predicted target genes were assessed in peripheral blood mononuclear cells from 46 patients with active pulmonary TB, 15 subjects with LTBI, and 17 non-infected healthy subjects (NIHS). THP-1 cell lines were transfected with miR-23a-3p mimics under stimuli with Mycobacterium TB-specific antigens. Both miR-155-5p and miR-150-5p gene expressions were decreased in the active TB group versus the NIHS group. Both miR-23a-3p and miR-146a-5p gene expressions were decreased in active TB patients with high bacterial burden versus those with low bacterial burden or control group (LTBI + NIHS). TLR2, TLR4, and interleukin (IL)10 gene expressions were all increased in active TB versus NIHS group. MiR-23a-3p mimic transfection reversed ESAT6-induced reduction of reactive oxygen species generation, and augmented ESAT6-induced late apoptosis and phagocytosis, in association with down-regulations of the predicted target genes, including tumor necrosis factor (TNF)-α, TLR4, TLR2, IL6, IL10, Notch1, IL6R, BCL2, TGF-β1, SP1, and IRF1. In conclusion, the down-regulation of miR-23a-3p in active TB patients with high bacterial burden inhibited mononuclear cell function and phagocytosis through TLR4/TNF-α/TGF-β1/IL-10 signaling via targeting IRF1/SP1.

## 1. Introduction

The incidence of active tuberculosis (TB) disease was estimated to be 10.4 million cases, with 1.7 million deaths during 2016. Rapid and accurate detection of active TB is essential for guiding treatment, yet case detection and reporting rates remain low, with 40% of estimated incident cases failing to be identified and reported. Thus, biomarkers are urgently required to detect active TB disease and differentiate it from latent TB infection (LTBI). Augmentation of beneficial immune responses, host-directed therapy, might serve as useful adjunct therapy to anti-TB drug-treatment regimens to shorten treatment duration, limit immunopathology by modulating aberrant *Mycobacterium TB* (Mtb)-induced immune responses, minimize permanent lung damage, and prevent new drug resistance [1,2].

Micro RNA (miRNAs; miR) are small non-coding RNAs that inhibit their target gene expressions and can affect host immunity to Mtb infection through the modulation of inflammatory cytokines and chemokines, such as tumor necrosis factor (TNF)-α, interleukin (IL)-6, and through stimulation of macrophage polarization. Evidence is emerging that suggests miRNAs could serve as diagnostic biomarkers for TB and anti-TB immunotherapy targets [3,4]. Previous studies have shown that toll-like receptor (TLR) 4 plays positive roles in the induction of immune responses against Mtb and in eradication of the infection, while TLR2 contributes to host protection, immune evasion and immune regulation during chronic Mtb infection [5,6]. MiR-21, miR-23a, miR-146a, and miR-150 all attenuate inflammatory pathways by down-regulating TLR2 or TLR4 expression, while miR-155 inhibits negative regulators of the TLR4/MyD88 signaling pathway [7,8,9,10,11]. However, the expression profiles and functions of these TLR-related miRNA in Mtb-induced lesions remain elusive [12].

Based on previous findings of the relationships between TLR2/4-mediated immune responses and active TB disease, it is hypothesized that patients with active pulmonary TB and subjects with LTBI may have aberrant expressions of the five miRNAs regulating TLR signaling, including miR-21-5p, miR-23a-3p, miR-146a-5p, miR-150-5p, and miR-155-5p, which may affect the development and clinical phenotypes of active TB disease. To test this hypothesis, this prospective cohort study checked the expressions of the five miRNAs and the predicted target genes in peripheral blood mononuclear cells (PBMCs) from patients with active TB disease, subjects with LTBI, and non-infected healthy subjects (NIHS). Furthermore, human monocytic cell lines were transfected with specific miRNA mimics under stimuli with M.tb-specific antigen.

## 2. Results

### 2.1. Demographics of the Participants 

A total of 78 subjects, including 46 patients with sputum culture positive pulmonary TB patients, 15 LTBI patients, and 17 NIHS, were enrolled and analyzed. Characteristics of cases and controls are listed in Table 1. The study population was all Asian in ethnicity. Age, smoking history, alcoholism history, and co-morbidity were all matched between the three groups, but the active TB group had a larger proportion of male patients than the other two groups. 

### 2.2. Decreased miR-155-5p, miR-150-5p, miR-23a-3p, and miR-146a-5p Gene Expressions in Active TB Patients, Especially in Those with High Bacterial Burden

MiR-155-5p gene expression levels were decreased both in the active TB group (0.67 ± 0.55 fold change, adjusted *p* = 0.027) and LTBI group (0.52 ± 0.38 fold change, adjusted *p* = 0.017) as compared with those in the NIHS group (1.31 ± 0.98 fold change, Figure 1a). MiR-150-5p gene expression levels were decreased in the active TB group (0.35 ± 0.32 fold change, Figure 1b), as compared with either LTBI subjects (0.78 ± 0.74 fold change, adjusted *p* = 0.002) or NIHS (0.95 ± 0.64 fold change, adjusted *p* < 0.001). Subgroup analysis showed that miR-23a-3p gene expression levels were decreased in active TB patients with high bacterial burden (defined as sputum AFB 2+ to 4+ at diagnosis, *n* = 17; 0.51 ± 0.25 fold change, Figure 1c) as compared with that in those with low bacterial burden (defined as sputum AFB 0 to 2+, *n* = 29; 1.16 ± 0.13 fold change, adjusted *p* = 0.009) or control groups (including LTBI subjects and NIHS, *n* = 32; 1.17 ± 0.64, adjusted *p* = 0.049). Gene expression levels of miR-146a-5p were decreased in active TB patients with high bacterial burden (0.31 ± 0.24 fold change, Figure 1d) as compared with that in those with low bacterial burden (2.46 ± 2.75 fold change, adjusted *p* = 0.008) or control groups (including LTBI subjects and NIHS, *n* = 32; 1.17 ± 0.85 fold change, adjusted *p* = 0.047). Additionally, miR-146a-5p gene expression level was further decreased in active TB patients with delayed response after 2-month anti-TB therapy (*n* = 30, 0.53 ± 0.4 fold change, Figure 1e) as compared with that in those with rapid response (*n* = 16, 1.45 ± 1.09 fold change, adjusted *p* = 0.01) or control groups (LTBI + NIHS, *n* = 32, 1.3 ± 0.83 fold change, adjusted *p* = 0.008). 

In 15 patients with active TB disease whose blood samples were obtained after 6-month anti-TB therapy, miR-21-5p gene expression levels were reduced after treatment as compared with that before treatment (0.46 ± 0.31 versus 1.22 ± 0.85 fold change, *p* = 0.005, Figure 1f). 

### 2.3. Decreased miR-23a-3p/miR-155-5p Gene Expression and Increased MiR-146a-5p Gene Expression in Response to ESAT-6 and CFP-10 Stimuli In Vitro

In vitro ESAT-6 and CFP-10 stimuli in human monocytic THP-1 cells resulted in down-regulations of both miR-23a-3p (*p* < 0.05, Figure 2a) and miR-155-5p genes (*p* < 0.05, Figure 2b), and up-regulations of the miR-146a-5p (*p* < 0.05, Figure 2c), TLR4 (*p* < 0.05, Figure 2d), TLR2 (*p* < 0.05, Figure 2e), and TNF-α (*p* < 0.05, Figure 2f) genes, but no significant change in either miR-21-5p or miR-150-5p gene expression. 

### 2.4. Predicted Target Genes of miR-23a-3p

Only miR-23a-3p and miR-155-5p were consistently down-regulated both in active TB patients and in response to Mtb-specific antigen stimuli in vitro, while miR-155 has been demonstrated to be able to eliminate mycobacterial infection through augmenting apoptosis, autophagy, and ROS production via targeting Rheb/SHIP1/TLR4 signaling in previous studies [13,14,15,16,17,18]. In contrast, little is known about the role of miR-23a-3p in mediating immunologic responses against M.tb, so miR-23a-3p was selected for further functional studies. To determine the target genes related to miR-23a-3p, the common targets and pathways of the miR-23a-3p regulated by M.tb were explored by the genes intersection option using the miRbase database. The results identified several miR-23-3p-regulated targets and pathways, some of which were involved in immunologic responses (five direct targets: IL6R, BCL2, TGF-β1, SP1, IRF1; six indirect targets: IL6, TNF-α, TLR4, TLR2, IL10, NOTCH1) and thus selected for further evaluation in both clinical samples and in vitro experiments. 

### 2.5. Increased Target Gene Expressions of miR-23a-3p, Including IL10, TLR4, and TLR2 Genes, in Active TB Patients

IL10 gene expression was increased in the active TB group as compared with that in the NIHS group (8.86 ± 17.2 versus 1.14 ± 1.08 fold change, adjusted *p* = 0.015, Figure 3a). IL10 gene expression was further increased in those with delayed response to 2-month anti-TB treatment (12.27 ± 20.23 fold change, *n* = 30, Figure 3b) as compared with that in either those with rapid response (0.95 ± 1.06 fold change, *n* = 16, adjusted *p* < 0.001) or control groups (NIHS + LTBI, *n* = 32; 1.62 ± 2.13 fold change, adjusted *p* = 0.016). TLR2 gene expression was increased in active TB group (3.57 ± 4.5 fold change, Figure 3c) as compared with that either in the LTBI (0.98 ± 1.19 fold change, adjusted *p* =0.031) or NIHS group (1.42 ± 1.38 fold change, adjusted *p* = 0.015). TLR4 gene expression was increased both in active TB (2.18 ± 2.91 fold change, adjusted *p* = 0.007) and LTBI (3.54 ± 3.42 fold change, adjusted *p* = 0.038) groups as compared with that in the NIHS group (0.98 ± 0.42 fold change, Figure 3d). MiR-23a-3p gene expression was negatively correlated with gene expression levels of the IL-10 (r = −0.408, *p* < 0.001, Figure 3e), IL6 (r = −0.466, *p* < 0.001, Figure 3f), BCL2 (r = −0.454, *p* < 0.001), TNF-α (r = −0.43, *p* < 0.001), NOTCH1 (r = −0.393, *p* < 0.001), SP1 (r = −0.393, *p* = 0.001), IRF1 (r = −0.354, *p* = 0.001), and TLR4 (r = −0.23, *p* = 0.047) genes. On the other hand, miR-23a-3p gene expression was positively correlated with TLR4 over the IL10 gene expression ratio (r = 0.393, *p* < 0.001, Figure 3g). 

In the 15 active TB patients with second blood sampling, TLR2 (2.18 ± 1.94 versus 4.87 ± 4.06 fold change, *p* = 0.003, Figure 3h) and BCL2 (1 ± 1.19 versus 3.65 ± 4.05 fold change, *p* = 0.018, Figure 3i) gene expressions were both reduced after 6-month anti-TB therapy as compared with that before treatment.

### 2.6. MiR-23a-3p Transfection in THP-1 Cells Resulted in Increased ROS Production, Late Apoptosis, and Phagocytosis in Association with the Reversal of ESAT6-Induced Up-Regulations of The IL6R, BCL2, TGF-β1, IRF1, SP1, IL6, TNF-α, NOTCH1, TLR4, and IL10 Genes 

To investigate the protective effect of miR-23a-3p on immunologic response against Mtb, we first evaluated whether miR-23a-3p mimic could augment bactericidal capacity of THP-1 cells. We found that ESAT6 plus miR-nonsense sequence stimuli resulted in decreased ROS production and increased late apoptosis as compared with miR-nonsense sequence alone control, while miR-23a-3p mimic transfection at a concentration of 50 nM resulted in increased ROS production (*p* < 0.05, Figure 4a) and augmented late apoptosis (*p* < 0.05, Figure 4b) under ESAT6 stimuli as compared with the ESAT6 plus miR-nonsense sequence condition. Next, we evaluated whether miR-23a-3p mimic could augment phagocytosis function of the THP-1 macrophage. We found that miR-23a-3p mimic transfection resulted in increased up-take of ESAT6 antigene-antibody-CF488A compound as compared with miR-nonsense sequence transfection (Figure 4c). Then, we evaluated whether miR-23a-3p mimic had inhibitory effects on the predicted target genes in THP-1 cells under ESAT-6 stimuli. We found that miR-23a-3p mimic transfection at a concentration of 50 nM resulted in increased miR-23a-3p gene expression (*p* < 0.05, Figure 4d), and decreased gene expressions of the IL6R (*p* < 0.05, Figure 4e), BCL2 (*p* < 0.05, Figure 4f), TGF-β1 (*p* < 0.05, Figure 4g), IRF1 (*p* < 0.05, Figure 4h), SP1 (*p* < 0.05, Figure 4i), IL6 (*p* < 0.05, Figure 4j), TNF-α (*p* < 0.05, Figure 4k), NOTCH1 (*p* < 0.05, Figure 4l), TLR4 (*p* < 0.05, Figure 4m), and IL10 (*p* < 0.05, Figure 4n) genes (all *p* values < 0.05) as compared with miR-nonsense sequence control. ESAT6 plus miR-nonsense sequence stimuli resulted in opposite effects on these gene expressions versus miR-nonsense sequence control, while miR-23a-3p mimic transfection reversed all the aberrant gene expressions induced by ESAT6. On the other hand, TLR4/IL10 expression ratio was decreased with ESAT6 stimuli and reversed to normal with miR-23a-3p mimic transfection (Figure 4o). The cell viability of THP-1 under the four conditions is shown in Figure 4p.

## 3. Discussion

Accumulating evidence has demonstrated that several miRNAs play important roles during Mtb infection mainly via fine tuning the immune signaling pathways and participating extensively in the complicated regulations of host-pathogen interactions. Among these pathways, both TLR and IL-10 pathways are very important and frequently involved. TLR2/4 plays positive roles in the induction of immune responses against Mtb and participates in eradication of the infection, while IL-10 can have deleterious effects on the patients during active TB disease in terms of bacterial clearance [19]. In the current study, we found that miR-155-5p, miR-150-5p, miR-23a-3p, and miR-146a-5p were all down-regulated in patients with active pulmonary disease with high bacterial burden, while only miR-155-5p and miR-23a-3p were down-regulated with Mtb-specific antigens stimuli in vitro. We further demonstrated that miR-23a-3p transfection could reverse the ESAT6-induced reduction of ROS generation and augment ESAT6-induced late apoptosis in THP-1 cells via targeting TNF-α/TLR4/IL10/TGF-β1/SP1/IRF1 signaling. The SP1 transcriptional regulatory element is present within the TLR4, TNF-α, and TGF-β1 gene promoters, while the IRF1 transcriptional regulatory element is present within the IL10 gene promoter [20,21,22,23]. Furthermore, SP1 and IRF1 have been shown to be direct targets of miR-23a in previous studies [24,25,26]. Thus, it is reasonable to infer that miR-23a-3p could mediate the imbalance between TLR4/TNF-α pro-inflammatory and IL10/TGF-β1 anti-inflammatory pathways under Mtb stimuli via targeting SP1 and IRF1 directly.

MiR-23a-27a-24 is located in the human chromosome 9q22, forming three mature miRNAs, and are involved in several biological processes of cancer, including proliferation, cell cycle arrest, apoptosis, differentiation, invasion, and metastasis [27]. Reduced levels of mature miR-23a in various tumors are primarily due to epigenetic silencing or alterations in biogenesis pathways, while inhibition of miR-23a in stressed cells represent a general mechanism for inducing apoptosis [28]. Deleting miR-23a in T cells upon acute Listeria monocytogenes infection could result in excessive inflammation, massive liver damage, and a marked mortality increase through boosting cytosolic ROS flux and breaking mitochondrial integrity [29]. The expression of miR-23a has been shown to be reduced in LPS-treated macrophages, causing suppression of NF-κB and pro-inflammatory genes, including IL-6 and TNF-α, through targeting A20 [30]. MiR-23a was reported to be down-regulated in Klebsiella pneumoniae-infected pulmonary epithelial cells and affect integrin function to regulate bacterial adhesion [31]. Both miR-23a-3p and miR-23a-5p were shown to be inhibited following infection with Aeromonas hydrophila in grass carp fish, repressing TNF-α/IFN/IL-8 and promoting caspase-3/7 by targeting CiGadd45ab [32]. Accordingly, miR-23a-3p down-regulation was documented both in active TB patients and in response to M.tb-specific antigen stimuli in vitro in the current study. Moreover, miR-23a-3p mimic transfection could reverse the immune inhibitory effects induced by ESAT-6 or augment apoptosis of immune cells. In contrast, one previous in vitro study found that miR-23a-5p could promote intracellular mycobacteria survival through inhibiting the activation of autophagy against Mtb by targeting TLR2/MyD88/NF-kappaB pathway [9]. One of the reasons for this discrepancy may be the difference between 5p and 3p arms of miRs, which are the two sides of the pre-miR stem-loop, and may have different sequences, targets, functions, and tissue-dependent expressions [33]. Additionally, Zhengjun Yi et al. reported high regulation of miR23a in sputum samples from TB patients [34]. When comparing the studies investigating miRNA levels in TB, it is important to interpret their results while taking into account their sample sizes, sample types, sampling time, and measuring methods. None of Yi’s and our studies had the recommended sample size of 70 cases and 110 controls. Sputum samples are easily collected and analyzed, but may be problematic as samples taken from patients can have varying amounts of cells in them. All blood samples had been collected before anti-TB drugs were given in our study, while sampling time was not described in Yi’s report. Finally, miR-23a expression levels were determined by the microarray platform in Yi’s study, while miR-23a under-expression was documented by quantitative Reverse-Transcriptase Polymerase Chain Reaction (RT-PCR) methods in active TB patients with high bacterial burden and in response to ESAT6 stimuli in vitro. Thus, miRNA-based diagnosis and therapy still have a long way to go before they can be applied in the clinical setting. Consensus amongst investigation groups should be sought to allow standardization of protocols and reproducibility of results. Previous studies have shown that reduced levels of mature miR-23a in various tumors, primarily due to epigenetic silencing or alterations in biogenesis pathway, represent a general mechanism for inducing apoptosis, whereas several studies have proposed opposite functions of the miR-23a, even when they studied the same type of cancer [28,35]. The role of miR-23a mimic in mycobacterial infection has yet been extensively investigated. Further challenging THP-1 cells with Mtb to examine intracellular bacterial viability in cells transfected with miR-23a-3p mimic is needed to clarify its potential bactericidal capacity.

Recent studies have shown that miR-150 has a crucial regulatory role in cancers. Interestingly, depending on the mRNA targeted by miR-150, it can act as either oncogenic miR or tumor suppressor in both malignant hematopoiesis and solid tumors. MiR-150 can regulate B cell/T cell/natural killer cell differentiation, augment the expression of inflammatory mediators in myeloid-specific Kruppel-like factor 2 knock-out macrophages, and contribute to the fibrosis process of systemic sclerosis. Moreover, miR-150 plays a key role in memory CD8 T cell differentiation through a c-Myb-controlled enhanced survival circuit and suppression of Foxo1 [36,37]. Mir-150 also plays a cell-intrinsic role in the regulation of effector CD8+ T cell fate and efficiency in killing cells infected by Listeria monocytogenes or vaccinia virus. Up-regulated circulatory miR-150 has been shown to be associated with poor outcomes of both influenza H1N1 infection and dengue fever [38,39]. In the current study, miR-150-5p gene expression was down-regulated in patients with active pulmonary TB disease. However, we did not find any significant change in miR-150-5p gene expression levels of monocytic THP-1 cell lines or PBMC samples from healthy subjects (data not shown) in response to Mtb-specific antigen stimuli in vitro. On the other hand, Sonic HH signaling-regulated miR-150 up-regulation induced by Mycobacterium bovis BCG has been shown to target MyD88, leading to the suppression of TLR2 responses in macrophages [40]. One reason for this discrepancy is that miR-150-induced response may be highly pathogen-and-cell-specific, as observed in several other miRs. It is also possible that other unknown molecules may be required to inhibit miR-150-5p expressions in addition to Mtb-specific antigen stimuli. Further investigations are needed to clarify underlying mechanisms by which Mtb reactivation leads to miR-150-5p suppression.

MiR-146a has been shown to facilitate bacterial replication in macrophage through inhibiting nitric oxide generation and reducing the induction of pro-inflammatory cytokines via targeting IRAK-1 and TRAF-6 [41,42]. MiR-146a was reported to be up-regulated in serum from adult pulmonary TB patients, but down-regulated both in alveolar macrophage from patients with active TB disease and in plasma from pediatric TB patients [43,44,45]. Over-expression of miR-146a was found to enhance the killing ability of THP-1 cells to intracellular M. bovis BCG via targeting PTGS2 [45]. In the current study, miR-146a-5p was down-regulated in active TB patients with high bacterial burden, but up-regulated in THP-1 cell with ESAT6/CFP10 stimuli. Similar to the findings regarding many other miRNAs, miR-146a appears to show divergent expressions and dual functions, depending on tissue types, virulence of pathogens, disease stages, and conditions of in vitro experiments [4].

Several limitations in the current study should be acknowledged. First, gene expression levels of the miRs and predicted target genes may be affected by several other clinical factors and the time points of blood samplings. However, we made a linear regression analytic model to minimize the effects of all potential confounding factors, such as age, gender, and co-morbidities. Second, most of the altered gene expressions in active TB patients at diagnosis did not return to normal after completing anti-TB therapy. One of the reasons may be the relatively small sample size in the paired comparison before and after treatment. In eight active TB patients with high bacterial burden and second blood samplings, gene expression levels of the miR-23a-3p and its target genes tend to become normal after treatment (data not shown). Another reason is that several TB survivors still have some form of persistent pulmonary dysfunction despite microbiologic cure [46]. Third, the effect of miR-23a-3p on the survival of live M.tb was not tested. However, we demonstrate that miR-23a-3p could bolster ROS generation and enhance late apoptosis of macrophage in line with the findings in previous studies, which may open the possibility of applying miR-23a-3p mimic to host-directed immunotherapy in TB [29,44]. The suggestion is supported by previous studies, which have shown that miR-23a is indispensable for effector CD4(+) T cell expansion, and that miR-23a and miR-27a promote apoptosis in human granulosa cells [29,47].

In conclusions, miR-155-5p and miR-23a-3p were both down-regulated in active TB patients and in response to Mtb-specific antigen stimuli. Previous sophisticated studies have demonstrated that miR-155 maintains the survival of both Mtb-infected macrophages and Mtb-specific T cells through SHIP1/AKT pathways, enabling an effective late adaptive immune response but dampening the early innate immune response [48]. Thus, the present study focuses on the role of miR-23a-3p in the regulation of the Mtb-induced immune response. The results suggest that there was a negative association between miR-23a-3p and TLR4/TNF-α/IL6/IL10/BCL2/SP1/IRF1 gene expressions. In active pulmonary TB patients with high bacterial burden, miR-23a-3p expression was decreased, affecting the balance between TLR4 and IL10 immunologic responses by down-regulating SP1/IRF1 expression and further influencing TLR4/IL10 expression ratio. Over-expression of miR-23a-3p resulted in elevated ROS generation, enhanced late apoptosis, and augmented phagocytosis in response to ESAT6 stimuli. These findings may contribute to the understanding of the role of miR-23a-3p in active TB disease and LTBI, and indicate that over-expression of the miR-23a-3p may be a new host-directed immunotherapy for active TB disease.

## 4. Materials and Methods

### 4.1. Study Subjects

The study population consisted of 46 patients with newly diagnosed pulmonary TB and 15 patients with newly diagnosed LTBI who were undergoing anti-TB or LTBI treatment at the Pulmonary Department of the Chang Gung Memorial Hospital (Kaohsiung, Taiwan) from January 2019 to September 2020. The criteria for the enrollment of active pulmonary TB were clinical and radiological findings, and at least 1 positive M.tb culture from sputum examinations or one bronchial washing specimen obtained by bronchoscopy. Patients with HIV or concomitant infection other than M.tb were excluded. Acid fast bacilli (AFB) smears and mycobacterial cultures were performed, and standard posterior-anterior chest X-ray (CXR) was assessed for disease severity at diagnosis, and after 2-/6-month anti-TB therapy. The criteria for enrollment of LTBI were the absence of pulmonary lesions on CXR, a negative history of TB disease, and positive for interferon-γ releasing assay. Seventeen NIHS were recruited, and the criteria for enrollment were the absence of pulmonary lesions on CXR examination, a negative history of TB disease, and negative for interferon-γ releasing assay test. The Chang Gung memorial hospital’s institutional review board approved the study protocol (certificate number: 201801482B0; approval date: 2018/10/11; Chang Gung Medical Foundation Institutional Review Board), and all subjects provided informed written consents before blood sampling. All patients were treated according to the American Thoracic and Infectious Society guidelines for the management of TB, and received directly observed treatment, short course strategy. Delayed response to treatment was defined as resolution of less than half the original lesions on CXR or positive sputum AFB smear/Mtb culture after 2-month anti-TB therapy.

### 4.2. Isolation of Leukocyte RNA and Protein from Blood Leukocyte Samples

Peripheral blood mononuclear cells were isolated from heparinized blood of all study subjects using a two-layer Ficoll-Histopaque density gradient centrifugation (Histopaque 1.077 and 1.119; Sigma Diagnostics, St. Louis, MO, USA) method. Blood samples at diagnosis before anti-TB treatment were obtained and analyzed from all study participants, and after six months of anti-TB treatment (month 6) from 15 active TB patients. Samples were stored in RNAlater^®^ RNA Stabilization Solution (Ambion^®^) at −80 °C until analysis. An RNeasy^®^ Plus Mini Kit (Qiagen, Hilden, Germany) was used for isolation of high quality total RNA, and treated with DNase according to the manufacture’s protocol.

### 4.3. Analysis of miR Gene Expressions

cDNA was generated from 2 µL of purified total RNA using the TaqMan Advanced miRNA cDNA Synthesis kit (Thermo Fisher Scientific, Waltham, MA, USA). Additionally, 1 pM of the synthetic C. Elegans oligo, cel-miR-39 (Sequence: UCACCGGGUGUAAAUCAGCUUG), was added to the isolated total RNA. This sequence does not exist in humans and was used as an exogenous control. All qPCR reactions were normalized to their corresponding cel-miR-39 Ct values. Quantitative RT-PCR was performed for each sample using 2.5 µL of diluted cDNA, TaqMan Advanced miRNA Assays (cel-miR-39-3p: 478293_mir; hsa-miR-150-5p: 477918_mir; hsa-miR-155-5p: 477927_mir; hsa-miR-146a-5p: 478399_mir; hsa-miR-21-5p: 477975_mir; hsa-miR-23a-3p: 478532_mir; Thermo Fisher Scientific, Waltham, MA, USA), and Applied Biosystems™ TaqMan™ Fast Advanced Master Mix (Thermo Fisher Scientific, Waltham, MA, USA) under fast cycling conditions. All TaqMan assay quantitative RT-PCR was carried out using the ABI 7500 fast Real-Time PCR System (Applied Biosystems, Foster City, CA, USA). Real-time PCR cycling conditions consisted of 95 °C for 20 s, followed by 40 cycles of 95 °C for 3 s and 60 °C for 30 s. Expression changes were determined by the 2-ΔΔCT method.

### 4.4. Analysis of Target Gene mRNA Expressions of Isolated PBMCs Using Quantitative Reverse-Transcriptase Polymerase Chain Reaction (RT-PCR)

To determine the expressions of the predicted target genes, the gene expressions of the TNF-α, TLR4, TLR2, IL6, IL10, NOTCH1, IL6R, BCL2, TGF-β1, SP1, and IRF1 genes were analyzed using quantitative RT-PCR in a 96-well format. The housekeeping gene GAPDH was chosen as an endogenous control to normalize the expression data for each gene. All PCR primers (random hexamers) were designed and purchased from Roche according to the company’s protocols (www.roche-applied-science.com), and their sequences are given in Table 2. RNA samples were treated with DNAfree to remove contaminating genomic DNA. A total of 300 ng RNA was used for synthesis of first strand cDNA with QuantiTectReverse Transcription Kit (QIAGEN, Hilden, Germany) with the reverse transcription reaction added to 5 μL of master mix (QIAGEN, SYBR Green PCR kit; Roche, Hilden, Germany). The PCR reactions with 45 cycles of amplification were run in a Roche CyclerQuantiFast R system, and a single real-time PCR experiment was carried out on each sample for each target gene. Relative expression levels were calculated using the ΔΔ*C*t method with the median value for the control group as the calibrator.

### 4.5. In Vitro Cell Culture Model under M.tb-Specific Antigen Stimuli

Human THP-1 monocytic cells (1 × 10^6^ cells) were seeded into a 96-well plate for 24 h, and stimulated with pre-specified concentrations of the recombinant proteins for 48 h: 5 μg/mL for 6 kDa early secretory antigenic target (ESAT-6, BEI resources, NIH, Bethesda, MD, USA) and 5/μg/mL for 10 kDa culture filtrate antigenic protein (CFP-10, BEI resources, NIH). All experiments were performed in quadruplicate, independently. Lipopolysaccharide (LPS) was used as a positive control. Wells left un-stimulated were the negative control.

### 4.6. Transfection of miRNA-23a-3p Mimic in THP-1 cells

MiR-23a-3p mimic (final concentration, 50 nM) and miR-nonsense sequence control (NC) were synthesized by GenePharma (Shanghai, China) and incubated in THP-1 cells with Lipofectamine 2000 (Invitrogen, Carlsbad, CA, USA) for 6 h to over-express the gene expression level of miR-23-5p using the HiPerFect transfection reagent (QIAGEN, Hilden, Germany).

### 4.7. Measurement of THP-1 Intracellular Reactive Oxygen Species (ROS)

A fresh stock solution of H2DCFDA (catalog no. D6883; Sigma, St. Louis, MO, USA) was diluted to make 0.1 μM working solution, which was then added to the THP-1 cells at a density of 1 × 106 cells/mL. Cell-associated mean fluorescent intensity was measured by flow cytometry in FL1 channel at excitation and emission wavelengths of 488 and 535 nm, respectively, using the CytomicsTM FC500 (Beckman Coulter, San Jose, CA, USA).

### 4.8. Measurement of Cell Apoptosis by Flow Cytometry Analysis

Following treatment, THP-1 cell apoptosis rates were evaluated by flow cytometry using the CFTM 488A Annexin V/Propidium iodide (PI) apoptosis detection kit (Biotium, Fremont, CA, USA). Percentage of Annexin V and Propidium iodide double positive cells (late apoptosis) was measured in the FL1 channel at excitation and emission wavelengths of 488 and 530 nm, respectively, using the CytomicsTM FC500 (Beckman Coulter).

### 4.9. Phagocytosis Assay

THP-1 cells were treated with 30 ng/mL phorbol-12-myristate 13-acetate (PMA, Sigma-Aldrich, Saint Luis, MO, USA) for 3 days, and induced into macrophage. EAST-6 antigen (1 μg/mL, BEI, Manassas, VA, USA) was incubated with antibody (EAST-6, Abcam, Massachusetts, USA) for 15 min, followed by incubation with dye (CF488A, Sigma-Aldrich, Saint Luis, MO, USA) for 15 min. THP-1 macrophage was then transfected (HiPerFect Transfection Reagent, Qiagen, MD, USA) with either nonsense sequence control (50 nM, Thermo Scientific, Waltham, MA, USA) or miR-23a-3p mimic (50 nM, Thermo Scientific) for 24 h, followed by treatment with ESAT6 antigen-antibody-CF488A compound. Phagocytosis was assayed by flowcytometry, using the CytomicsTM FC500 (Beckman Coulter). Specific ESAT6 antigen-antibody-CF488A up-take of each sample was expressed as a percentage of CF-488A (+) cells.

### 4.10. Measurement of Cell Viability (Mitochondrial Activity) by WST-1

WST-1 reagent (Roche, Mannheim, Germany) diluted 1:10 in growth medium was added into THP-1 cells grown in a 96-well plate (10^4^ cells/well). The amount of viable cells was determined via optical density measurement using a microplate reader at 450 nm, with 600 nm as a reference wavelength.

### 4.11. Statistical Analysis

Continuous values were expressed as mean ± standard deviation. The differences between groups were analyzed by ANOVA, paired t, Kruskal–Wallis H, Mann–Whitney U, and chi-square tests, where appropriate. Multiple linear regression analysis was used to minimize the effects of confounding factors on the subgroup comparisons of continuous variables, and to provide adjusted *p* values and 95% confidence intervals (CI). Correlations between two continuous variables were evaluated using Spearman’s correlation coefficient. The null hypothesis was rejected at *p* < 0.05. Analyses were performed using the SPSS 22.0 statistical software (SPSS Corp., Chicago, IL, USA).

## 5. Conclusions

In conclusions, miR-155-5p and miR-23a-3p were both down-regulated in active TB patients and in response to Mtb-specific antigen stimuli. Previous sophisticated studies have demonstrated that miR-155 maintains the survival of both Mtb-infected macrophages and Mtb-specific T cells through SHIP1/AKT pathways, enabling an effective late adaptive immune response but dampening the early innate immune response [45]. Thus, the present study focuses on the role of miR-23a-3p in the regulation of the Mtb-induced immune response. The results suggested that there was a negative association between miR-23a-3p and TLR4/TNF-α/IL6/IL10/BCL2/SP1/IRF1 gene expressions. In active pulmonary TB patients with high bacterial burden, miR-23a-3p expression was decreased, affecting the balance between TLR4 and IL10 immunologic responses by down-regulating SP1/IRF1 expression and further influencing TLR4/IL10 expression ratio. Over-expression of miR-23a-3p resulted in elevated ROS generation, enhanced late apoptosis, and augmented phagocytosis in response to ESAT6 stimuli. These findings may contribute to the understanding of the role of miR-23a-3p in active TB disease and LTBI, and indicate that over-expression of the miR-23a-3p may be a new host-directed immunotherapy for active TB disease. More evidence and research are needed before the miR-23a-3p mimics can be used in clinical applications.

## Figures and Tables

**Figure 1 ijms-21-08587-f001:**
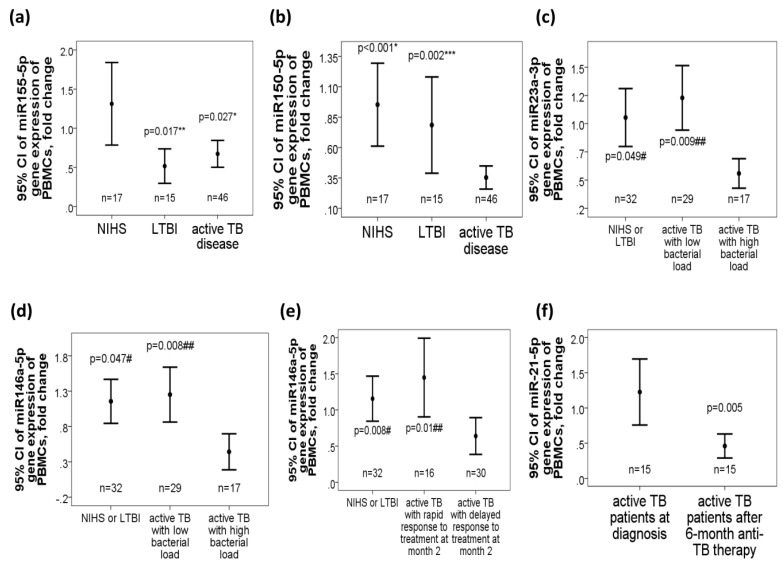
Differential gene expressions of the five microRNAs targeting the toll-like receptor (TLR) pathway in tuberculosis (TB) patients. (**a**) MiR-155-5p gene expression was decreased in patients with active pulmonary TB disease and latent TB infection (LTBI) versus non-infected healthy subjects (NIHS). (**b**) MiR-150-5p gene expression was decreased in the active TB group versus either the LTBI or NIHS group. Gene expressions of both (**c**) miR-23a-3p and (**d**) miR-146a-5p were decreased in active TB patients with high bacterial load versus either those with low bacterial load or NIHS. (**e**) Mir-146a-5p gene expression was further decreased in active TB patients with delayed response to 2-month anti-TB therapy versus either those with rapid response or control group (LTBI + NIHS). (**f**) MiR-21-5p gene expression was reduced in active TB patients after 6-month anti-TB therapy versus before therapy. * *p* < 0.05, compared between active TB patients and NIHS (adjusted by linear regression analysis model) ** *p* < 0.05, compared between LTBI and NIHS (adjusted by linear regression analysis model) *** *p* < 0.05, compared between active TB patients and LTBI (adjusted by linear regression analysis model) # *p* < 0.05, compared between NIHS and active TB patients with delayed response (adjusted by linear regression analysis model) ## *p* < 0.05, compared between active TB patients with delayed response and those with rapid response (adjusted by linear regression analysis model).

**Figure 2 ijms-21-08587-f002:**
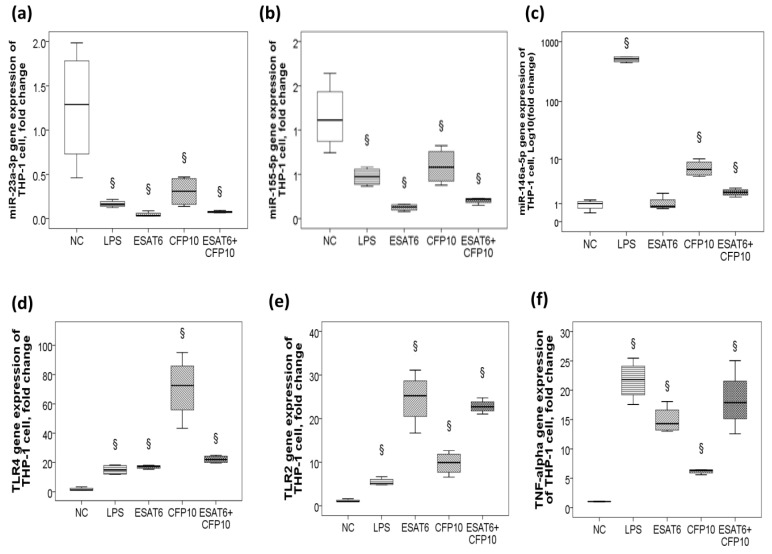
Differential gene expressions of the five microRNAs targeting the toll-like receptor pathway in response to Mycobacterium TB-specific antigen stimuli in vitro. In vitro ESAT6, CFP10, and lipopolysaccharide (LPS) stimuli in THP-1 cells resulted in down-regulations of (**a**) miR-23a-3p and (**b**) miR-155-5p, and up-regulations of (**c**) miR-146a-5p, (**d**) TLR4, (**e**) TLR2, and (**f**) tumor necrosis factor (TNF)-α. § *p* < 0.05, compared with normal control (NC; culture medium) by Kruskal–Wallis H test.

**Figure 3 ijms-21-08587-f003:**
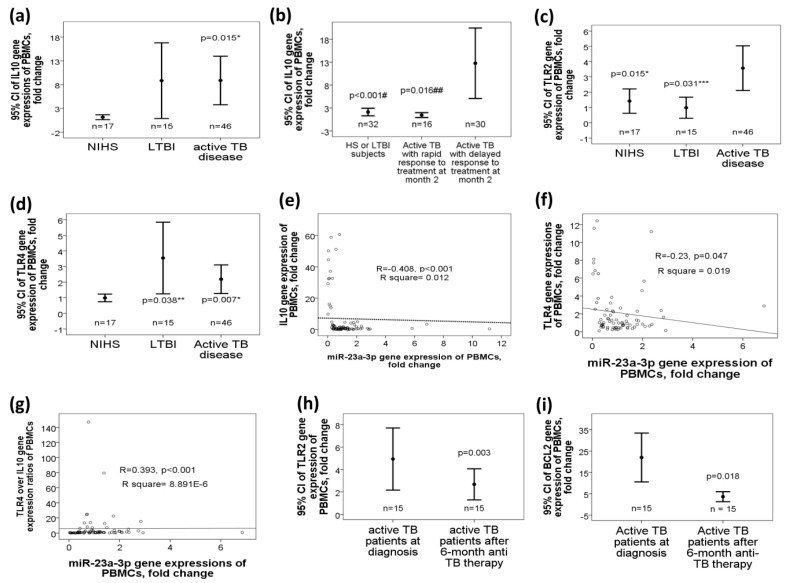
Differential gene expressions of the predicted target genes of miR-23a-3p in TB patients, and the effect of miR-23a-3p mimic transfection on macrophage functions. Interleukin (IL) 10 gene expression was increased in (**a**) active TB patients versus NIHS, and further increased in that of (**b**) those with delayed response to 2-month anti-TB therapy versus those with rapid response or the control group (LTBI + NIHS). (**c**) TLR2 gene expression was increased in active TB versus either the LTBI or NIHS group. (**d**) TLR4 gene expression was increased both in active TB and LTBI groups versus NIHS. MiR-23a-3p gene expression was negatively correlated with (**e**) IL10 and (**f**) TLR4 gene expressions, and (**g**) positively correlated with TLR4/IL10 gene expression ratios. Both (**h**) TLR2 and (**i**) BCL2 gene expressions were reduced after 6-month anti-TB therapy. * *p* < 0.05, compared between active TB patients and NIHS (adjusted by linear regression analysis model) ** *p* < 0.05, compared between LTBI and NIHS (adjusted by linear regression analysis model) *** *p* < 0.05, compared between active TB patients and LTBI (adjusted by linear regression analysis model) # *p* < 0.05, compared between NIHS and active TB patients with delayed response (adjusted by linear regression analysis model) ## *p* < 0.05, compared between active TB patients with delayed response and those with rapid response (adjusted by linear regression analysis model)

**Figure 4 ijms-21-08587-f004:**
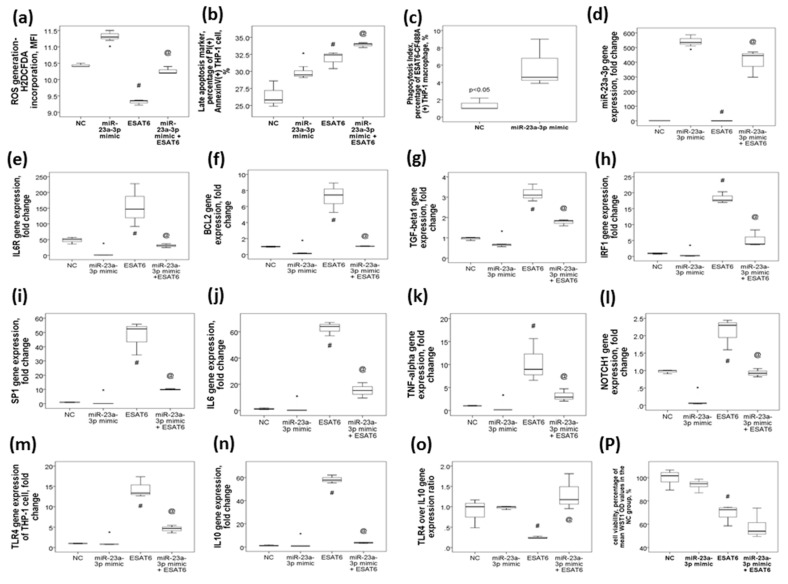
The effect of miR-23a-3p mimic transfection on monocyte/macrophage function and gene expressions of the miR-23a-3p and its target genes under the stimuli of ESAT6 in THP-1 cells. Transfection with miR-23a-3p mimic in THP-1 cells followed by ESAT6 stimuli for 24 h resulted in (**a**) increased generation of reactive oxygen species, (**b**) increased late apoptosis, and (**c**) increased phagocytosis versus nonsense sequence control (NC) plus the ESAT6 condition. ESAT6 stimuli resulted in the down-regulation of (**d**) mi-23a-3p and up-regulations of its predicted target genes, including (**e**) IL6R, (**f**) BCL2, (**g**) TGF-β1, (**h**) IRF1, (**i**) SP1, (**j**) IL6, (**k**) TNF-α, (**l**) NOTCH1, (**m**) TLR4, and (**n**) IL10, while miR-23a-3p transfection reversed all of these altered gene expressions. (**o**) TLR4/IL10 gene expression ratios were decreased with ESAT6 stimuli and reversed to normal with miR-23a-3p transfection. (**p**) Viability of THP-1 cells was determined by the percentage of mean WST1 incorporation in the NC group. * *p* < 0.05, compared between miR-nonsense sequence control (NC) and miR-23a-3p mimic transfection by Kruskal–Wallis H test # *p* < 0.05, compared between NC and ESAT6 plus NC by Kruskal–Wallis H test @ *p* < 0.05, compared between ESAT6 plus NC and miR-23a-3p transfection plus ESAT6 by Kruskal–Wallis H test.

**Table 1 ijms-21-08587-t001:** Demographic, biochemistry, and clinical data of all the 78 study participants.

	Non-Infected Healthy Subjects *N* = 17	Subjects with Latent TB Infection *N* = 15	Patients with Active Pulmonary TB Disease *N* = 46	*p* Value
Age, years	53.86 ± 17.73	59 ± 10.55	59.7 ± 14.77	0.428
Male sex, n (%)	5 (35.7)	7 (46.7)	32 (69.6)	0.012
Co-morbidity, n (%)				
Hypertension	5 (29.4)	4 (26.7)	9 (19.6)	0.666
Diabetes mellitus	2 (11.8)	2 (13.3)	16 (34.8)	0.085
COPD/Asthma	0 (0)	1 (6.7)	6 (13)	0.258
Chronic hepatitis	2 (11.8)	1 (6.7)	8 (17.4)	0.557
Chronic kidney disease	2 (11.8)	1 (6.7)	2 (4.3)	0.565
Heart failure	2 (11.8)	1 (6.7)	2 (4.3)	0.565
Alcoholism, n (%)	0 (0)	0 (0)	5 (10.9)	0.196
Current Smoker, n (%)	3 (17.6)	2 (13.3)	14 (30.4)	0.312
Sputum smear at diagnosis, n (%)				
Acid fast bacilli 0			20 (43.5)	
Acid fast bacilli 1+			9 (19.6)	
Acid fast bacilli 2+			4 (8.7)	
Acid fast bacilli 3+			5 (10.9)	
Acid fast bacilli 4+			8 (17.4)	
Drug-resistant TB, n (%)			4 (8.6)	
CXR at diagnosis, n (%)				
Far advanced lesions			20 (43.5)	
Minimal to moderate			26 (56.5)	
Pleural effusion			9 (19.6)	
Delayed resolution after 2-month anti-TB therapy			25 (54.3)	
Systemic symptoms, n (%)			14 (30.4)	
Fever			10 (21.7)	
Body weight loss			7 (15.2)	

COPD = chronic obstructive pulmonary disease.

**Table 2 ijms-21-08587-t002:** Primer sequences used in the quantitative reverse-transcriptase polymerase chain reaction experiments.

Gene Name	Primer	Sequence
*TLR2*	forward	5′- CGTTCTCTCAGGTGACTGCTC -3′
	reverse	5′- CCTTTGGATCCTGCTTGC-3
*TLR4*	forward	5′-TGGAAGTTGAACGAATGGAATGTG-3′
	reverse	5′-ACCAGAACTGCTACAACAGATACT-3′
*TNF-* *α*	forward	5′- CCCCAGGGACCTCTCTCTAA-3′
	reverse	5′CTCAGCTTGAGGGTTTGCTAC-3′
*IL6R*	forward	5′- AGCCTCCCAGTGCAAGATTC-3′
	reverse	5′- GGTATTGTCAGACCCCAGGC-3′
*IRF1*	forward	5′- CATCCCGCCTGAACTTG -3′
	reverse	5′- AGGCTGGTCTCGAACTC -3′
*SP1*	forward	5′- CGACCTTCTGGTATCTTGT-3′
	reverse	5′- CTTGCTTTCTATCAGATCAGGG-3′
*IL10*	forward	5′-GCTGGAGGACTTTAAGGGTTACCT-3′
	reverse	5′-CTTGATGTCTGGGTCTTGGTTCT-3′
*TGF-* *β1*	forward	5′-CAAGGGCTACCATGCCAACT-3′
	reverse	5′-AGGGCCAGGACCTTGCTG-3′
*NOTCH1*	forward	5′-GAGGCGTGGCAGACTATGC-3′
	reverse	5′-CTTGTACTCCGTCAGCGTGA-3′
*IL6*	forward	5′-ACTCACCTCTTCAGAACGAATTG-3′
	reverse	5′-CCATCTTTGGAAGGTTCAGGTTG-3′
*BCL2*	forward	5′-TTGTGGCCTTCTTTGAGTTCGGTG-3′
	reverse	5′-GGTGCCGGTTCAGGTACTCAGTCA-3′

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
