# Peer review of "MicroRNA-23a-3p Down-Regulation in Active Pulmonary Tuberculosis Patients with High Bacterial Burden Inhibits Mononuclear Cell Function and Phagocytosis through TLR4/TNF-α/TGF-β1/IL-10 Signaling via Targeting IRF1/SP1"

_ijms, 2020, doi:10.3390/ijms21228587_

Round 1
Reviewer 1 Report
The paper by Young-Che-Chen et al. is dedicated to the verification of the hypothesis that patients with active TB may have disturbed expression of the five miRNAs regulating TLR signaling, which may affect the development and clinical phenotypes of active TB disease. The topic is interesting, however manuscript requires some corrections.
Abstract
Lines 35-37: The text says “ down regulation of miR-23a-3p in active TB patients with high bacterial burden inhibited bactericidal capacity of macrophage through TLR4/TNF-a/TGF-b1/IL-10 signaling via targeting IRF1/SP1”. Contrary to this, Authors did not investigate the bactericidal capacity of macrophages in TB patients. The results obtained in the THP-1 model only suggest this possibility.
Materials and Methods
4.1. Study subjects
Lines 306-310: I would like to see more information characterising the LTBI and NIHS groups. Were these volunteers family members of patients with newly diagnosed tuberculosis? If so, some volunteers negative for interferon-gamma releasing assay test (the NIHS group) may have eradicated M.tb earlier than developed adaptive immunity. Reversal of the results of this test occurs frequently in contacts of TB patients. Druszczynska M. et al. Indian J Microbiol. 56(2):205-213.
Discussion
Lines 287-289: Authors should consider a possibility that “most of the altered gene expression in active TB patients at diagnosis did not return to normal after completing anti-TB therapy” because several TB survivors have some form of persistent pulmonary dysfunction despite microbiologic cure (Ravimohan S, Kornfeld H, Weissman D, et al. Eur Respir Rev 2018; 27: 170077)
Lines 292-295: Conclusions should be carefully stated and moderated. The authors inform that the effect of miR-23a-3p on the survival of live M.tb was not tested. Nevertheless, they write that miR-23a-3p could booster ROS generation and enhance late apoptosis of macrophage (THP-1), which may open the possibility of applying miR-23a-3p mimic to host-directed immunotherapy in TB. The suggestion is supported by other authors who have shown that miR-23a is indispensable for effector CD4(+) T cell expansion (29) and miR-23a and miR-27a promote apoptosis in human granulose cells (44).
Author Response
Response to Reviewer 1
The paper by Young-Che-Chen et al. is dedicated to the verification of the hypothesis that patients with active TB may have disturbed expression of the five miRNAs regulating TLR signaling, which may affect the development and clinical phenotypes of active TB disease. The topic is interesting, however manuscript requires some corrections.
Abstract
Lines 35-37: The text says “ down regulation of miR-23a-3p in active TB patients with high bacterial burden inhibited bactericidal capacity of macrophage through TLR4/TNF-a/TGF-b1/IL-10 signaling via targeting IRF1/SP1”. Contrary to this, Authors did not investigate the bactericidal capacity of macrophages in TB patients. The results obtained in the THP-1 model only suggest this possibility.
Ans.: Thank you for your comments. As suggest, we made a change to the conclusions and title as follows: “In conclusions, down-regulation of miR-23a-3p in active TB patients with high bacterial burden inhibited bactericidal capacity of macrophage mononuclear cell function and phagocytosis through TLR4/TNF-α/TGF-β1/IL-10 signaling via targeting IRF1/SP1.”
Materials and Methods
4.1. Study subjects
Lines 306-310: I would like to see more information characterising the LTBI and NIHS groups. Were these volunteers family members of patients with newly diagnosed tuberculosis? If so, some volunteers negative for interferon-gamma releasing assay test (the NIHS group) may have eradicated M.tb earlier than developed adaptive immunity. Reversal of the results of this test occurs frequently in contacts of TB patients. Druszczynska M. et al. Indian J Microbiol. 56(2):205-213.
Ans.: All subjects in the LTBI and NIHS groups had a history of contact with patients with active TB disease, but were not relatives or family members of the active TB group.
Discussion
Lines 287-289: Authors should consider a possibility that “most of the altered gene expression in active TB patients at diagnosis did not return to normal after completing anti-TB therapy” because several TB survivors have some form of persistent pulmonary dysfunction despite microbiologic cure (Ravimohan S, Kornfeld H, Weissman D, et al. Eur Respir Rev 2018; 27: 170077)
Ans.: As suggest, we add this comment into the discussion section.
Lines 292-295: Conclusions should be carefully stated and moderated. The authors inform that the effect of miR-23a-3p on the survival of live M.tb was not tested. Nevertheless, they write that miR-23a-3p could booster ROS generation and enhance late apoptosis of macrophage (THP-1), which may open the possibility of applying miR-23a-3p mimic to host-directed immunotherapy in TB. The suggestion is supported by other authors who have shown that miR-23a is indispensable for effector CD4(+) T cell expansion (29) and miR-23a and miR-27a promote apoptosis in human granulose cells (44).
Ans.: As suggest, we moderate the conclusions as follows: “More evidence and research are needed before the miR-23a-3p mimics can be used in clinical applications.”
Reviewer 2 Report
Chen et al seek to investigate the role of miRNA in active tuberculosis disease and LTBI, particularly miRNA targeting toll-like receptor pathway signalling. The authors describe miRNA gene expression of 5 selected miRNA in PBMCs isolated from patients with active TB, LTBI or healthy controls.
Results
Figure 1 - the authors report differential expression of five candidate miRNAs in NIHS, LTBI and Tb disease graphed as 95% CI. The raw data of these results should be provided. Can the authors show the ΔΔCT values for the complete data set rather than 95% CI. Did the authors consider an endogenous reference gene for normalisation?
This figure is too densely populated and is unclear for reader. I would suggest making two figures – one with miR data from PBMCs and another with data from cell line.
Was any difference seen in the expression of miR23 or miR146a between NIHS and LTBI? Similarly with miR21 - a microRNA that has shown to be of significant importance in mycobacterial infection – was any difference seen between NIHS and LTBI, only miR21 expression data from active disease is shown. Barry et al 2018 showed downregulation of miR21 in sputum of active patients compared to non-infected individuals. It would be interesting to see how miR21 expression compares across the three cohorts in this study.
Yi et al 2012 reported high regulation of miR23a in sputum samples from TB patients – can the authors comment how they might interpret in the context of their findings?
Axis legends should have minimal text, cell type should be in figure legend rather than axis legend.
Statistical symbols of (g) – (l) not represented in figure legend.
The authors report miR155 decreased in THP-1 cells upon stimulation with ESAT6 and CFP10. Bonilla-Muro et al 2019 showed upregulation of miR155 in MDMS stimulated with rExsA – can the authors comment in relation to their study?
Did authors check cell viability of THP-1 cells stimulated with Mtb antigens? ESAT6 has been reported to induce apoptosis? Please demonstrate cell viability in your assay
Figure 2 – again the reviewer requests raw data for ΔΔCT analysis rather than 95% CI.
Section 2.5 refers to TLR4 as Figure 2c whereas the figure has TLR2 as Figure 2c, similar for Figure 2d and related text.
Please explain the statistical logic and significance of analysing TLR4 over IL-10 gene expression?
Figure 2j,k an dl are discussed in section 2.6 with results from Figure 3 – authors should considering merging the figures for this section.
In evaluation of “whether miR-23a-3p mimic could augment bactericidal capacity of THP-1 cells” did the authors consider challenging THP-1 cells with Mtb to examine intracellular bacterial viability in cells with miR-23a-3p mimic? This would support the hypothesis put forward by the authors.
Fig2k – axis is % and should start at 0%.
It is unclear in Fig 2j-l if cells treated with ESAT6 have been treated with miR-nonsense sequence and are comparable to NC – please clarify
Timepoint in should be included in figure legend for ESAT6 stimulation
Figure 3 - Axis legends should have minimal text, cell type should be in figure legend rather than axis legend.
Relevance and interpretation of Fig3l is questionable.
It is unclear in Fig 3 if cells treated with ESAT6 have been treated with miR-nonsense sequence and are comparable to NC – please clarify.
Can authors clarify that Figure 3 represent data analysis using 2−ΔΔCT to calculate fold change as graphs are labelled.
Timepoint in should be included in figure legend for ESAT6 stimulation
In order to support the title of this manuscript the authors need to demonstrate inhibition of bactericidal capacity of macrophages by microRNA-23a-3p, ideally by colony forming units.
The authors chose to focus on miR-23a-3p to investigate the functional role during mycobacterial infection, however much of the discussion focuses on the other miRs analysed in Figure 1. Further discussion on the potential role and function of miR-23a-3p in bacterial infection would be relevant.
For the phagocytosis assay THP-1 cells were treated with PMA to mature into macrophages, however for the other assays – Mtb antigen stimulation, ROS assay, transfection mimics, Flow cytometry – THP-1 cells were not treated with PMA and thus were monocytes. Can the authors please comment on this and explain why a monocytic cell line was chosen yet for the phagocytosis assay macrophages were used.
General comment – figures need to be reformatted
Author Response
Response to Reviewer 2
Chen et al seek to investigate the role of miRNA in active tuberculosis disease and LTBI, particularly miRNA targeting toll-like receptor pathway signalling. The authors describe miRNA gene expression of 5 selected miRNA in PBMCs isolated from patients with active TB, LTBI or healthy controls.
Results
Figure 1 - the authors report differential expression of five candidate miRNAs in NIHS, LTBI and Tb disease graphed as 95% CI. The raw data of these results should be provided. Can the authors show the ΔΔCT values for the complete data set rather than 95% CI. Did the authors consider an endogenous reference gene for normalisation?
Ans.: Thank you for your comments. Fold change (mean and standard deviation) for each group has been given in the text, while the ΔΔCT values are listed below.
|
Gene |
miR-21 |
miR-23a |
miR-146a |
miR-150 |
miR-155 |
|
ΔΔCT of NIHS group |
0.1283±0.6472 |
0.2748±1.0456 |
0.1427±1.0588 |
0.5844±1.5202 |
0.2488±1.3616 |
|
ΔΔCT of LTBI group |
0.3106±0.4019 |
0.644±1.2355 |
0.2395±0.7653 |
1.4525±2.1861 |
1.2583±1.4802 |
|
ΔΔCT of active TB group |
0.6263±0.7719 |
0.4587±1.0432 |
0.6208±1.1901 |
1.6084±1.228 |
0.9106±1.3811 |
Qiagen recommends that cel-miR-39-3p could be an ideal spike in miRNA control for even serum samples with relatively high PCR efficiency but it has to be added only after incubation with lysis reagent.
This figure is too densely populated and is unclear for reader. I would suggest making two figures – one with miR data from PBMCs and another with data from cell line.
Ans.: As suggest, we separate the old Figure 1 into 2 figures.
Was any difference seen in the expression of miR23 or miR146a between NIHS and LTBI? Similarly with miR21 - a microRNA that has shown to be of significant importance in mycobacterial infection – was any difference seen between NIHS and LTBI, only miR21 expression data from active disease is shown. Barry et al 2018 showed downregulation of miR21 in sputum of active patients compared to non-infected individuals. It would be interesting to see how miR21 expression compares across the three cohorts in this study.
Yi et al 2012 reported high regulation of miR23a in sputum samples from TB patients – can the authors comment how they might interpret in the context of their findings?
Ans.: There was no significant difference in miR-23a, miR146a, or miR21 gene expression between NIHS and LTBI group. We could not find the paper of Yi et al 2012 via PubMed search. miR gene expression levels may change with disease severity, immune response course, sample types, time of sampling, etc. We just report the findings in this cohort.
Axis legends should have minimal text, cell type should be in figure legend rather than axis legend.
Statistical symbols of (g) – (l) not represented in figure legend.
Ans.: As suggest, we correct these errors.
The authors report miR155 decreased in THP-1 cells upon stimulation with ESAT6 and CFP10. Bonilla-Muro et al 2019 showed upregulation of miR155 in MDMS stimulated with rExsA – can the authors comment in relation to their study?
Ans.: Rothchild, A. C. e al 2016 showed that miR-155 maintains the survival of Mtb-infected macrophages, thereby providing a niche favoring bacterial replication; on the other hand, miR-155 promotes the survival and function of Mtb-specific T cells, enabling an effective adaptive immune response. Wagh, V. et al 2017 reported that the levels of miR-155 were significantly reduced in serum of TB patients as compared to uninfected controls. Thus, we speculate that miR-155 changes in response to Mtb-specific antigen stimuli may vary depending on the cell type used in the experiment, antigen types, treatment duration, and the presence of other immune cells.
Did authors check cell viability of THP-1 cells stimulated with Mtb antigens? ESAT6 has been reported to induce apoptosis? Please demonstrate cell viability in your assay
Ans.: We add the data of THP-1 cell viability in new Figure 4p.
Figure 2 – again the reviewer requests raw data for ΔΔCT analysis rather than 95% CI.
Ans.: TheΔΔCT values of each group are shown below.
|
|
TLR2 |
TLR4 |
IL10 |
|
ΔΔCT of NIHS group |
-0.0216±1.3453 |
0.1207±0.4875 |
-0.0214±0.8539 |
|
ΔΔCT of LTBI group |
-1.1275±2.4646 |
-0.896±0.8751 |
-0.8457±2.0984 |
|
ΔΔCT of active TB group |
-0.0784±1.8143 |
-0.3042±1.0742 |
-0.4787±2.066 |
Section 2.5 refers to TLR4 as Figure 2c whereas the figure has TLR2 as Figure 2c, similar for Figure 2d and related text.
Ans.: As suggest, we correct these errors.
Please explain the statistical logic and significance of analysing TLR4 over IL-10 gene expression?
Ans.: We add the following limitation in the discussion section. “TLR4 is a major innate immune response against Mtb infection, while IL-10 functions as an anti-inflammation response favoring Mtb survival. Thus, we speculate TLR4/IL-10 ratio may serve as a marker reflecting positive and negative balance between inflammatory and anti-inflammatory response. However, more research is required to prove this theory.”
Figure 2j,k an dl are discussed in section 2.6 with results from Figure 3 – authors should considering merging the figures for this section.
Ans.: As suggest, we merge these figures into new Figure 4.
In evaluation of “whether miR-23a-3p mimic could augment bactericidal capacity of THP-1 cells” did the authors consider challenging THP-1 cells with Mtb to examine intracellular bacterial viability in cells with miR-23a-3p mimic? This would support the hypothesis put forward by the authors.
Ans.: Challenging THP-1 cell with Mtb requires Biosafety level III laboratory, which is not equipped in our hospital. We change the title into “MicroRNA-23a-3p down-regulation in active pulmonary tuberculosis patients with high bacterial burden inhibits mononuclear cell function and phagocytosis through TLR4/TNF-α/TGF-β1/IL-10 signaling via targeting IRF1/SP1”.
Fig2k – axis is % and should start at 0%.
It is unclear in Fig 2j-l if cells treated with ESAT6 have been treated with miR-nonsense sequence and are comparable to NC – please clarify
Timepoint in should be included in figure legend for ESAT6 stimulation
Ans.: The THP-1 cells treated with ESAT6 had been treated with miR-nonsense sequence. We clarify if in the new Figure 4 legend.
Figure 3 - Axis legends should have minimal text, cell type should be in figure legend rather than axis legend.
Ans.: There are two types of cells tested in the current study, PBMCs and THP-1 cell. For clarity, we reserve the cell type in axis legends.
Relevance and interpretation of Fig3l is questionable.
Ans.: As mentioned above, TLR4 is a major innate immune response against Mtb infection, while IL-10 functions as an anti-inflammation response favoring Mtb survival. Thus, we speculate TLR4/IL-10 ratio may serve as a marker reflecting positive and negative balance between inflammatory and anti-inflammatory response.
It is unclear in Fig 3 if cells treated with ESAT6 have been treated with miR-nonsense sequence and are comparable to NC – please clarify.
Can authors clarify that Figure 3 represent data analysis using 2−ΔΔCT to calculate fold change as graphs are labelled.
Ans.: The old Figure 3 represents gene expression changes by using 2−ΔΔCT to calculate fold change.
Timepoint in should be included in figure legend for ESAT6 stimulation
In order to support the title of this manuscript the authors need to demonstrate inhibition of bactericidal capacity of macrophages by microRNA-23a-3p, ideally by colony forming units.
Ans.: As mentioned above, challenging THP-1 cell with Mtb requires Biosafety level III laboratory, which is not equipped in our hospital. We change the title into “MicroRNA-23a-3p down-regulation in active pulmonary tuberculosis patients with high bacterial burden inhibits mononuclear cell function and phagocytosis through TLR4/TNF-α/TGF-β1/IL-10 signaling via targeting IRF1/SP1”.
The authors chose to focus on miR-23a-3p to investigate the functional role during mycobacterial infection, however much of the discussion focuses on the other miRs analysed in Figure 1. Further discussion on the potential role and function of miR-23a-3p in bacterial infection would be relevant.
Ans.: As suggest, we add the following statement in the discussion section. “Previous studies have shown that reduced levels of mature miR-23a in various tumors, primarily due to epigenetic silencing or alterations in biogenesis pathway, represent a general mechanism for inducing apoptosis, whereas several studies have proposed opposite functions of the miR-23a, even when they studied the same type of cancer[28, 34]. The role of miR-23a mimic in mycobacterial infection has yet been extensively investigated. Further challenging THP-1 cells with Mtb to examine intracellular bacterial viability in cells transfected with miR-23a-3p mimic is needed to clarify its potential bactericidal capacity.”
For the phagocytosis assay THP-1 cells were treated with PMA to mature into macrophages, however for the other assays – Mtb antigen stimulation, ROS assay, transfection mimics, Flow cytometry – THP-1 cells were not treated with PMA and thus were monocytes. Can the authors please comment on this and explain why a monocytic cell line was chosen yet for the phagocytosis assay macrophages were used.
Ans.: Monocyte has less phagocytosis capacity than macrophage. We ever tested phagocytosis function of THP-1 monocyte, but found little change with either non-sense sequence or miR-23 mimic transfection. Only after transformation into macrophage was the significant phagocytosis observed.
General comment – figures need to be reformatted
Ans.: As suggest, the figures are reformatted.
Round 2
Reviewer 2 Report
Figure 1 - the authors report differential expression of five candidate miRNAs in NIHS, LTBI and Tb disease graphed as 95% CI. The raw data of these results should be provided. Can the authors show the ΔΔCT values for the complete data set rather than 95% CI. Did the authors consider an endogenous reference gene for normalisation?
Ans.: Thank you for your comments. Fold change (mean and standard deviation) for each group has been given in the text, while the ΔΔCT values are listed below.
The ΔΔCT values were given as a table with no stats accompanying. The fold change values given in the text refer to the data presented as 95% CI. miR data throughout the manuscript should be presented as the raw values from ΔΔCT analysis and not 95% CI fold change as data can be misinterpreted in this format.
Was any difference seen in the expression of miR23 or miR146a between NIHS and LTBI? Similarly with miR21 - a microRNA that has shown to be of significant importance in mycobacterial infection – was any difference seen between NIHS and LTBI, only miR21 expression data from active disease is shown. Barry et al 2018 showed downregulation of miR21 in sputum of active patients compared to non-infected individuals. It would be interesting to see how miR21 expression compares across the three cohorts in this study.
Yi et al 2012 reported high regulation of miR23a in sputum samples from TB patients – can the authors comment how they might interpret in the context of their findings?
Ans.: There was no significant difference in miR-23a, miR146a, or miR21 gene expression between NIHS and LTBI group. We could not find the paper of Yi et al 2012 via PubMed search.
Yi Z, Fu Y, Ji R, Li R, Guan Z. Altered microRNA signatures in sputum of patients with active pulmonary tuberculosis. PLoS One. 2012;7(8):e43184. doi: 10.1371/journal.pone.0043184. Epub 2012 Aug 10. PMID: 22900099; PMCID: PMC3416796
Figure 2 – again the reviewer requests raw data for ΔΔCT analysis rather than 95% CI.
miR data throughout the manuscript should be presented as the fold change values from ΔΔCT analysis and not 95% CI fold change as data can be misinterpreted/misleading in this format.
Fig2k – axis is % and should start at 0%.
This has not been amended
Axis legends should have minimal text, cell type should be in figure legend rather than axis legend.
This has not been amended
Figure 4 – This figure is very data heavy and difficult to navigate. The axis legends are much too small to read. This figure needs to be amended.
Author Response
Response to Reviewer 2_2nd round
Figure 1 - the authors report differential expression of five candidate miRNAs in NIHS, LTBI and Tb disease graphed as 95% CI. The raw data of these results should be provided. Can the authors show the ΔΔCT values for the complete data set rather than 95% CI. Did the authors consider an endogenous reference gene for normalisation?
Ans.: Thank you for your comments. Fold change (mean and standard deviation) for each group has been given in the text, while the ΔΔCT values are listed below.
The ΔΔCT values were given as a table with no stats accompanying. The fold change values given in the text refer to the data presented as 95% CI. miR data throughout the manuscript should be presented as the raw values from ΔΔCT analysis and not 95% CI fold change as data can be misinterpreted in this format.
Ans. (2nd round): Thank you for your comments. Fold change values have been adopted as a gold standard method for evaluation and comparisons of various gene expression levels, including miRs. The relative expression of each miRNA is calculated by the 2−ΔΔCq method and normalized on global mean or median of the control. (Scientific Reports (2019) 9:1584. BMC Medical Genomics 2013, 6(Suppl 1):S14)
Was any difference seen in the expression of miR23 or miR146a between NIHS and LTBI? Similarly with miR21 - a microRNA that has shown to be of significant importance in mycobacterial infection – was any difference seen between NIHS and LTBI, only miR21 expression data from active disease is shown. Barry et al 2018 showed downregulation of miR21 in sputum of active patients compared to non-infected individuals. It would be interesting to see how miR21 expression compares across the three cohorts in this study.
Yi et al 2012 reported high regulation of miR23a in sputum samples from TB patients – can the authors comment how they might interpret in the context of their findings?
Ans.: There was no significant difference in miR-23a, miR146a, or miR21 gene expression between NIHS and LTBI group. We could not find the paper of Yi et al 2012 via PubMed search.
Yi Z, Fu Y, Ji R, Li R, Guan Z. Altered microRNA signatures in sputum of patients with active pulmonary tuberculosis. PLoS One. 2012;7(8):e43184. doi: 10.1371/journal.pone.0043184. Epub 2012 Aug 10. PMID: 22900099; PMCID: PMC3416796
Ans. (2nd round): As suggest, we add the following comment in the discussion section.
“In contrast to our findings, Zhengjun Yi et al reported high regulation of miR23a in sputum samples from TB patients. When comparing the studies investigating miRNA levels in TB, it is important to interpret their results while taking into account their sample sizes, sample types, sampling time, and measuring methods. None of Yi’s and our studies had the recommended sample size of 70 cases and 110 controls. Sputum samples are easily collected and analyzed, but may be problematic as samples taken from patients can have varying amount of cells in them. All blood samples had been collected before anti-TB drugs were given in our study, while sampling time was not described in Yi’s report. Finally, miR-23a expression levels were determined by the microarray platform in Yi’s study, while miR-23a under-expression was documented by quantitative RT-PCR methods in active TB patients with high bacterial burden and in response to ESAT6 stimuli in vitro. Thus, miRNA-based diagnosis and therapy still have a long way to go before they can be applied in the clinical setting. Consensus amongst investigation groups should be sought to allow standardization of protocols and reproducibility of results.”
Figure 2 – again the reviewer requests raw data for ΔΔCT analysis rather than 95% CI.
miR data throughout the manuscript should be presented as the fold change values from ΔΔCT analysis and not 95% CI fold change as data can be misinterpreted/misleading in this format.
Ans. (2nd round): Fold change values have been adopted as a gold standard method for evaluation of various gene expression levels, including miRs. The relative expression of each miRNA is calculated by the 2−ΔΔCq method and normalized on global mean or median of the control. (Scientific Reports (2019) 9:1584. BMC Medical Genomics 2013, 6(Suppl 1):S14)
Fig2k – axis is % and should start at 0%.
Ans. (2nd round): All the percentage values of late apoptosis were more than 25 and far above 0, so the Y-axis started at 22.5%.
This has not been amended
Axis legends should have minimal text, cell type should be in figure legend rather than axis legend.
This has not been amended
Figure 4 – This figure is very data heavy and difficult to navigate. The axis legends are much too small to read. This figure needs to be amended.